# Left Ventricular Ejection Fraction Can Predict Atrial Thrombosis Even in Non-High-Risk Individuals with Atrial Fibrillation

**DOI:** 10.3390/jcm11143965

**Published:** 2022-07-07

**Authors:** Petra Angebrandt Belošević, Anton Šmalcelj, Nikola Kos, Krešimir Kordić, Karlo Golubić

**Affiliations:** 1Department of Cardiovascular Diseases, University Hospital Center Zagreb, 10000 Zagreb, Croatia; petraang37@yahoo.com; 2School of Medicine, University of Zagreb, 10000 Zagreb, Croatia; asmalcelj@gmail.com; 3Department of Cardiovascular Diseases, University Hospital Center “Sisters of Mercy”, 10000 Zagreb, Croatia; nikolakos89@gmail.com (N.K.); kresimir.kordic@yahoo.com (K.K.)

**Keywords:** atrial fibrillation, left atrial thrombus, ejection fraction, prediction model

## Abstract

Background—Current guidelines do not recommend routine use of transesophageal echocardiography (TOE) in anticoagulated patients with atrial fibrillation (AF). The aim of our study was to identify predictors for left atrial thrombosis (LAT) in patients with AF that would require TOE despite anticoagulation therapy, using clinical, laboratory and echocardiographic data which are usually obtained in those patients in a real-world setting. Methods—We analyzed data from electronic medical records (EMR) of consecutive AF patients referred to two university hospitals between January 2014 and December 2017 for pulmonary vein isolation (PVI) or direct current cardioversion. The primary endpoint was the presence of left atrial thrombus on TOE. Multivariable and univariable logistic regression models were computed using variables that were significantly different between the LAT and the control groups. Results—A total of 838 patients were included, of whom 132 (15.8%) had LAT. After controlling for other variables, only the left ventricle ejection fraction (LVEF) remained statistically significant with an OR of 0.956 (95% CI 0.934–0.979), *p* < 0.01. Regression models including LVEF had significantly higher areas under the receiver operating characteristic (ROC) curves, including in subgroups with non-high thromboembolic risk (CHA2DS2-Vasc = 0 or 1), with an area under the curve (AUC) of 0.76 (95% CI 0.71–0.81), *p* < 0.0001. Conclusions—The LVEF is an independent predictor of LAT, and it might improve thromboembolic risk stratification in future models. LVEF significantly increased the predictive value of the CHA2DS2-Vasc model and was able to identify LAT in non-high-risk patients.

## 1. Introduction

Transesophageal echocardiography (TOE) is often used to exclude left atrial thrombosis (LAT) in patients with atrial fibrillation (AF) undergoing cardioversion or pulmonary vein isolation (PVI). There are multiple reports of LAT in those patients despite anticoagulation therapy which poses a threat of cardioembolism [1]. Current guidelines do not recommend routine use of TOE in anticoagulated patients [2,3,4]. Currently, the CHA2DS2-VASc score is recommended for thromboembolic risk stratification and establishing indications for oral anticoagulation in patients with AF [4]. The aim of our study was to determine predictors of LAT using clinical, laboratory and echocardiographic data which are usually obtained in AF patients in a real-world setting in order to identify patients that would require TOE despite anticoagulation therapy and are otherwise low risk (CHA2DS2-Vasc = 0 or 1). In the end, we would compare our prediction model to CHA2DS2-VASc.

## 2. Materials and Methods

### 2.1. Patient Population

We included data from electronic medical records (EMR) of consecutive AF patients referred to two university cardiology departments between January 2014 and December 2017 for PVI or direct current cardioversion. All patients had TOE and transthoracic echocardiography (TTE) performed prior to the procedure. Patients were included in this study regardless of the presence or type of anticoagulation prior to TOE. The research protocol and the review of EMRs were approved by ethics committees of the Medical University of Zagreb, the University hospital center “Sisters of Mercy”, and the University Hospital Centre Zagreb. As this was an observational study involving only previously collected data, obtaining informed consent from the patients is not required by the committees nor under EU or national law.

### 2.2. Data Collection

Data derived from the EMRs included demographic data (sex and age), relevant medical history (hypertension, diabetes mellitus (DM), stroke/transient ischemic attack (TIA)/thromboembolism, vascular disease (prior myocardial infarction (MI), peripheral artery disease (PAD), or aortic plaque, and congestive heart failure), smoking habits (yes/no) and current medication (any and all antiaggregation and anticoagulation therapy). The CHA2DS2-Vasc score [4] was calculated from relevant variables. The following laboratory markers were used: hematocrit (HCT), platelet count, mean platelet volume (MPV), C-reactive protein (CRP), fibrinogen concentration, and the International Normalized Ratio (INR). Transthoracic echocardiography data included the left ventricle ejection fraction (LVEF) and left atrial dilatation as a binary variable. Because left atrial dimensions were not consistently reported in the same manor (usually volume was used, but sometimes also area and diameter), patients were divided into two groups (normal and dilated atria) in accordance with the Guidelines for Chamber Quantification of the European Association of Cardiovascular Imaging [5].

#### 2.2.1. Transesophageal Echocardiography

All patients scheduled for direct current cardioversion of AF or for PVI, irrespective of the presence or type of anticoagulation, have TOE and TTE performed routinely prior to the procedure. TOE examinations are usually conducted within a few hours prior to the scheduled procedure (at most within 24 h before the procedure). All TOE and TTE studies were performed by certified echocardiographers (accreditation in echocardiography of the Section of Echocardiography of the Croatian Cardiac Society), using the VIVID 9 Ultrasound Machine (General Electric, Boston, MA, USA). Images of selected patients are shown in the Appendix A. In doubtful cases, the study was evaluated by two echocardiographers, to establish a unanimous and most reliable diagnosis and to enable safe referral for ablation or cardioversion. Written informed consent for TOE was obtained from all patients.

#### 2.2.2. Study Endpoint

The primary endpoint was the presence of left atrial thrombus on TOE. We also recorded the presence of spontaneous echocontrast (SEC).

#### 2.2.3. Sample Size

The sample size was calculated using the formula for ROC curves of prediction models [6] under the following assumptions: alpha 0.05, beta 0.20, AUC 0.6, null hypothesis AUC 0.5, and a negative to positive ratio of 10. The result was 814 patients in total. Patients with missing data from certain variables were excluded from analysis.

#### 2.2.4. Statistical Analysis

Continuous data were presented as the median and the IQR. Group comparisons were conducted using the Chi-square test (with Yates’ correction where applicable) for quantitative variables, the *t*-test for normally distributed variables and the Mann–Whitney U test for non-normally distributed variables (normality of distribution was checked with the Kolmogorov–Smirnov test).

To check for the independence of predictors of LAT, univariable and multivariable logistic regression analyses were performed. Multivariable logistic regression models included all variables found to be predictors of LAT in univariable analyses, maintaining an adequate event per predictor variable value, with the addition of non-interacting already known predictors. ROC curves were constructed, and AUC was calculated to compare the sensitivity and specificity of different models. The CHA2DS2-Vasc model was used as a reference for comparison. Pairwise comparison of ROC curves was performed using the DeLong method [7]. Because of the exploratory nature of the group comparisons, Bonferroni’s adjustment for multiple hypothesis testing was applied yielding a *p*-value of less than 0.003 to be considered significant. For all other (pre-specified) tests, a *p*-value of less than 0.05 was considered significant. All tests were two tailed. All calculations were performed using SPSS version 20 (IBM, Armonk, NY, USA) MedCalc version 19.2.6 (MedCalc Software, Ostend, Belgium) and Office Excel 2007 (Microsoft, Redmond, Washington, DC, USA).

## 3. Results

### 3.1. Patient Characteristics

A total of 838 patients were included, of whom 132 (15.8%) had LAT, predominantly thrombosis of the left atrial (LA) appendage. The group differences are described in Table 1.

There were no significant differences in the recorded demographic variables. CRP was significantly higher in the LAT group. Elevated fibrinogen levels did not reach statistical significance. Other laboratory values also did not differ significantly.

### 3.2. Comorbidities and Clinical Thromboembolic Risk

Relative frequencies of CHA2DS2-Vascscore groups are shown superimposed in the Nightingale plot in Figure 1.

Most patients regardless of LAT were in the high-risk group (≥2), namely 475 (56.9%). There was no significant difference between the LAT and the control group; Yates’ chi-square: 10.69, *p*-value 0.22.

### 3.3. Antithrombotic Therapy

There were no significant differences between the two groups in either antiaggregation or anticoagulant therapy. There was also no significant difference in an unadjusted comparison between the vitamin K antagonists (VKA) and NOAC subgroups in LAT prevalence (78/494 vs. 16/106, respectively, *p* = 0.858). There was a typical pattern of LAT in patients using VKA according to INR with a steady decline toward the therapeutic values (2–3) and no benefit above 3.5. A significant number of patients taking VKA were outside of therapeutic INR values (44.4%).

### 3.4. Echocardiographic Data

LA enlargement frequency was not statistically different between the two groups. LVEF, on the other hand, was markedly higher in the control group, reaching statistical significance even with Bonferroni’s adjustment (*p*-value < 0.0001).

### 3.5. Logistic Regression Models and ROC Curves

One multivariable and five univariable logistic regression models were computed and are described in Table 2.

After controlling for other variables, only the EF remained statistically significant with an OR of 0.956 (0.951 in the univariate analysis). There was no significant interaction between the variables. Cox and Snell R2 was 0.06 (0.042 in the univariable analysis for EF).

In order to assess whether combining CHA2DS2-Vasc with LVEF would yield a significantly better prediction model, ROC curves were drawn, analyzed, and compared for the univariable LVEF model, CHA2DS2-Vasc model and the CHA2DS2-Vasc with LVEF model. The model including the LVEF was significantly better (AUC 0.59 vs. 0.69, *p* = 0.0015), as shown in Figure 2.

We also explored how EF would predict LAT in non-high-risk patients (that would, if no intervention was planned, usually not be anticoagulated), i.e., CHA2DS2-Vasc 0 or 1. The univariable EF model had an AUC of 0.76 (95% CI 0.71–0.81), *p* < 0.0001, with the cut-off being 57%, and Youden index J 0.495, as shown in Figure 3.

## 4. Discussion

The prevalence of LAT in our population was somewhat higher compared to other studies [1,8]. The reason could be common VKA administration with low time in therapeutic range (TTR), patient non-compliance with therapy, a high share of patients without any antithrombotic therapy or simply higher thromboembolic risk. The effect of demographic variables such as gender and age on LAT is too low to be visible in a population of this size, especially when adjusted for other variables.

Elevated markers of inflammation and procoagulant states are frequently encountered in LAT [9]. Although CRP was elevated in the LAT group and predicted LAT in a univariable regression model, it failed to show independence when adjusted for other variables. The CHA2DS2-Vasc score has been used as a predictor of LAT in AF, but with modest results [10]. We detected no significant differences between the LAT and control groups probably because the constituent components of the score were similar in both of them.

It is important to point out that the lack of detectable differences in antithrombotic therapies does not imply that the therapies were ineffective. The lack may be related to the heterogeneity of the population and different inherent thromboembolic risk. Our results regarding VKA and NOAC therapies agree with published data [8].

LA enlargement has been shown to predict LAT [11,12,13]. It also correlates well with the natural progression of AF from paroxysmal to persistent and then to permanent and reflects the amount of atrial fibrosis [14]. In our study LA enlargement also predicted LAT, but not independently after adjustment for other variables. The LVEF was reduced in the LAT group. This is in line with published data as a low LVEF is considered a risk factor for both stroke and death even in the absence of AF [15,16,17,18]. The LVEF has also in some studies been shown to be a predictor of LAT in different populations, but with no measures of predictive value [19,20]. Studies evaluating the predictive value of the LVEF were focused on negative predictive value and concluded that patients with normal LVEF and no other evaluated risk factors were at very low risk of LAT [8,21].

According to our results, of all examined variables, only LVEF seems to independently predict LAT in patients with AF. When added to CHA2DS2-Vasc in a logistic regression model, LVEF significantly improved prediction of LAT. Furthermore, in a non-high-risk population (CHA2DS2-Vasc 0 or 1), LVEF proved to be a fairly good predictor of LAT.

### Strengths and Limitations of This Study

To our knowledge, this is the first study examining the predictive value of the LVEF in regard to LAT with adjustments for not only clinical and echocardiographic risk factors, but also for biochemical markers of inflammation and coagulation at the same time. The strengths of this study include a large sample size, quality data, and a conscientious statistical analysis.

There are several limitations of our study. Firstly, the primary endpoint was the presence of left atrial thrombus on TOE, (which is a surrogate endpoint) and not ischemic stroke. However, LAA thrombus formation is considered the primary mechanism responsible for thromboembolic events in patients with AF and so seems appropriate to be used in risk estimation in AF. There are several sources of bias inherent to observational research, these include selection bias which limits generalizability, as our database included only hospitalized patients in only two centers. Additionally, as this is an observational study, we are limited to the sampled variables, more potential predictors might have gone unnoticed. On the other hand, the aim of this study was to identify risk factors that are routinely evaluated in clinical practice. The logistic regression models were not validated. Our study population was heterogeneous and included patients regardless of anticoagulation status; thus, the results of our analysis describe the combination of inherent and residual risk of thrombus formation despite oral anticoagulation.

## 5. Conclusions

In light of already published data, our study provides further evidence of the thrombogenic nature of the impaired left ventricular systolic function itself. As the LVEF is an independent predictor of LAT, it might improve thromboembolic risk stratification in future models. It is also an independent predictor able to identify LAT in non-high-risk patients (CHA2DS2-Vasc 0 or 1) with AF.

## Figures and Tables

**Figure 1 jcm-11-03965-f001:**
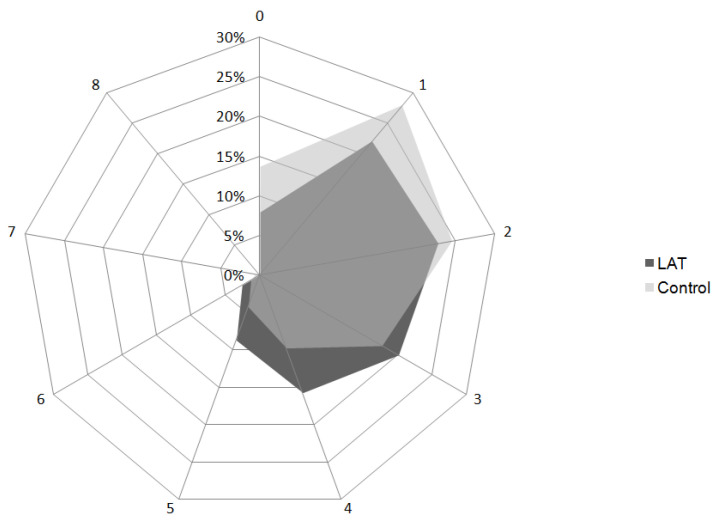
Nightingale plot of relative frequencies of CHA2DS2-Vasc score groups.

**Figure 2 jcm-11-03965-f002:**
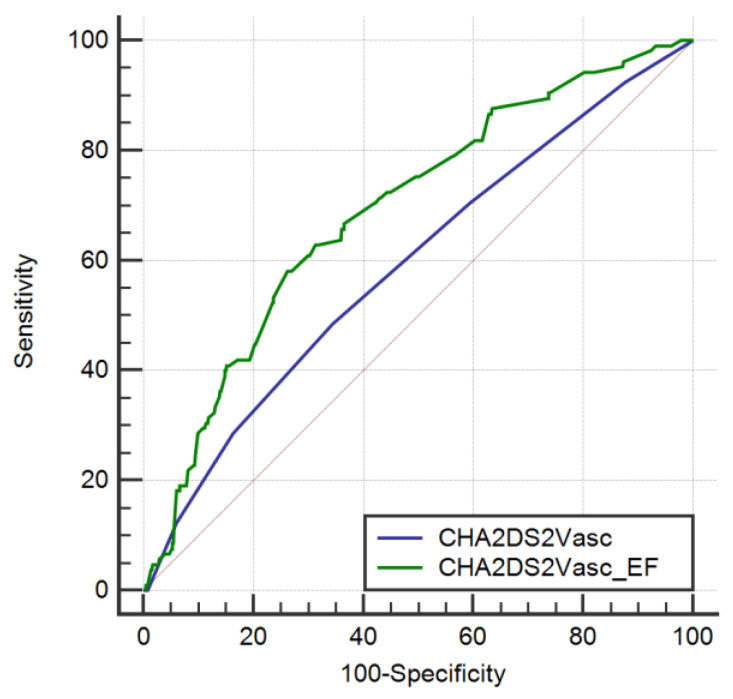
Comparison of ROC curves between the CHA2DS2-Vasc model and CHA2DS2-Vasc + LVEF model. The red line denotes the area of 0.5 of a random classifier.

**Figure 3 jcm-11-03965-f003:**
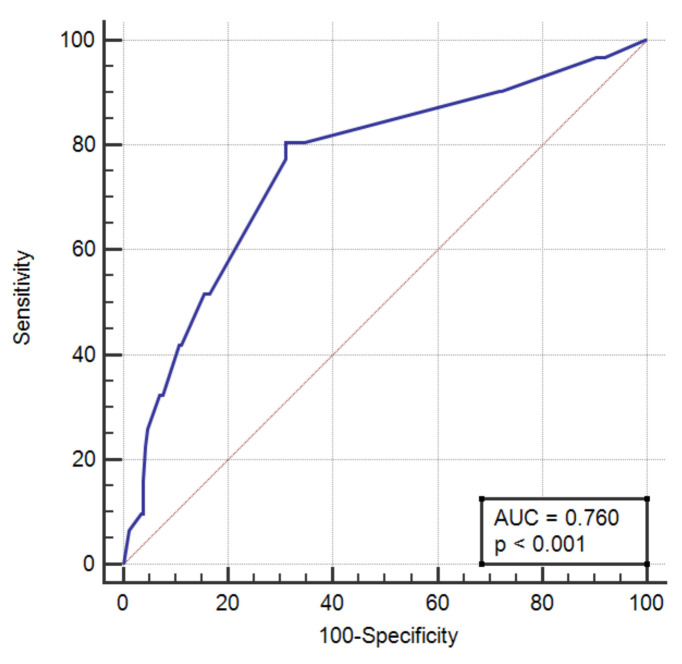
The ROC curve of the univariable LVEF model in patients with CHA2DS2-Vasc < 2.

**Table 1 jcm-11-03965-t001:** The difference between the LAT group and the control group regarding clinical, laboratory and echocardiographic variables recorded.

	LAT	IQR	%	Control	IQR	%	*p*
N	132		15.8	703		84.2	
Age (years)	65	58–70		63	56–70		0.067
Male sex (N)	92		69.7	494		70.3	0.896
SEC (N)	81		61.4	125		17.8	0
LA enlargement (N)	109		82.6	514		73.1	0.022
EF (%)	50	40–60		60	55–65		<0.0001
fibrinogen (g/L)	4.2	3.5–4.9		3.6	3.1–4.6		0.005
CRP (mg/L)	3.1	1.7–6.2		2.1	1.1–4.5		0.0003
HCT (L/L)	0.44	0.41–0.46		0.43	0.41–0.46		0.920
Thrombocytes (×10^9^/L)	197.5	173–241		207	177–238		0.289
MPV (FL)	9.1	8.6–9.8		9.2	8.6–9.9		0.920
CHA2DS2-Vasc							0.220
HA (N)	84		63.6	475		67.6	0.378
DM (N)	21		15.9	94		13.4	0.441
TIA (N)	6		4.5	33		4.7	0.938
PAD (N)	18		13.6	71		10.1	0.229
Antiaggregation (N)	24		18.2	122		17.4	0.818
VKA (N)	78		59.1	416		59.2	0.909
NOAC (N)	16		12.1	90		12.8	0.828
Smoking (N)	50		37.9	241		34.3	0.426
Symptomatic HF (N)	38		28.8	82		11.7	<0.0001
CAD (N)	22		16.7	79		11.2	0.079

SEC—spontaneous echo contrast; LA—left atrial; LVEF—left ventricle ejection fraction; CRP—C-reactive protein; HCT—hematocrit; MPV—mean platelet volume; AH—arterial hypertension; DM—diabetes mellitus; TIA—transient ischemic attack; PAD—peripheral artery disease; VKA—vitamin K antagonist; NOAC—novel oral anticoagulants; HF—heart failure; CAD—coronary artery disease.

**Table 2 jcm-11-03965-t002:** Multivariable and univariable logistic regression models.

Univariable	OR	95% CI		*p*	Cox and Snell R^2^
		Lower	Upper		
LA enlargement	3.605	1.636	7.942	<0.01	0.018
EF	0.951	0.935	0.967	<0.01	0.042
Fibrinogen	1.285	1.076	1.536	<0.01	0.015
CRP	1.012	0.999	1.026	0.08	0.003
Symptomatic HF	3.108	1.996	4.838	<0.01	0.027
Multivariable					0.06
Age	1.007	0.977	1.038	0.638	
Male sex	0.574	0.313	1.053	0.073	
LA enlargement	2.612	0.891	7.658	0.080	
EF	0.956	0.934	0.979	<0.01	
Fibrinogen	1.160	0.925	1.456	0.200	
CRP	1.002	0.978	1.026	0.876	
HA	0.904	0.450	1.813	0.776	
DM	1.445	0.696	3.002	0.323	
TIA	0.867	0.272	2.761	0.810	
PAD	1.646	0.621	4.364	0.316	
CAD	0.685	0.259	1.811	0.446	
NOAC	1.607	0.3	8.606	0.579	
VKA	0.819	0.401	1.673	0.583	

LA—left atrial; LVEF—left ventricle ejection fraction, CRP—C reactive protein; HF—heart failure; AH—arterial hypertension; DM- diabetes mellitus; TIA—transient ischemic attack; PAD—peripheral artery disease; CAD—coronary artery disease; NOAC—novel oral anticoagulants; VKA—vitamin K antagonist.

## Data Availability

The datasets used and/or analyzed during the current study are available from the corresponding author on reasonable request.

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
