# Peer review of "Left Ventricular Ejection Fraction Can Predict Atrial Thrombosis Even in Non-High-Risk Individuals with Atrial Fibrillation"

_jcm, 2022, doi:10.3390/jcm11143965_

Round 1
Reviewer 1 Report
In this study, the authors reported that left ventricle ejection fraction (LVEF) was an independent factor in left atrial thrombosis (LAT) in patients with atrial fibrillation (AF), and LVEF proved to be a fairly good predictor of LAT in a non-high-risk population (CHA2DS2-Vasc 0 or 1).
This study is very interesting and the manuscript is well written.
I have a few questions
# What was the percentage of paroxysmal AF (PAF) and non-PAF in the LAT and control groups?
# Warfarin and novel oral anticoagulants (NOAC) have been evaluated in combination. Has LVEF become an independent factor in LAT when the patients on NOAC alone were evaluated?
Author Response
Thank you for your kind review.
- We did not include the type of atrial fibrillation (paroxysmal, persistent, permanent) in our analysis.
- LVEF was an independent predictor of LAT when NOAC alone was evaluated (without VKA) in the multivariable model. There were no significant changes in the model parameters. However we could not perform the same multivariable analysis using only patients on NOAC as the logistic regression model failed to converge, probably due to significantly lower sample size.
Reviewer 2 Report
Dear editor;
In this study, In this study, Belošević et al. studied on 838 patients with AF who were receiving anticoagulation therapy. They aimed to identify predictors for left atrial thrombosis (LAT) in these patients using clinical, laboratory and echocardiographic data.The authors found LAT in 132 (15.8%) of the patients, predominantly with thrombosis of the left atrial (LA) appendix. It was noted that only the left ventricle ejection fraction (LVEF) was significant for LAT in study patients, when added to CHA2DS2-Vasc score in the logistic regression model.
As stated by the authors the prevalence of LAT in their population seemed to be higher compared to other studies.
Comments:
Although it is well written paper, I have some concerns;
1-The study is valuable as it evaluates LAT not only for clinical and echocardiographic risk factors, but also for biochemical markers of inflammation and coagulation.
2-The most important limitation of the study seems to be that it included only two centers and inpatients.
3-What about genetic or acquired thrombophilic factors between the groups?
4-It is recommended to share ECHO images showing LAT and EF in selected patients.
5-Please provide full explanations for CHA2DS2-Vasc, TOE and PVI or LAT etc. in the main text.
6-Please correct numbering in References.
Yours sincerely
Author Response
Thank you for your review.
3. There were no known genetic or acquired thrombophilic factors in the studied patients.
4. We have added ECHO images showing LAT and EF in selected patients.
5. Explanations for CHA2DS2-Vasc, TOE and PVI or LAT etc. have been added to the main text.
6. The numbering in References has been corrected.